# Graphite Nodularity Evaluation in High-Si Ductile Cast Irons

**DOI:** 10.3390/ma15217685

**Published:** 2022-11-01

**Authors:** Iulian Riposan, Denisa Anca, Iuliana Stan, Mihai Chisamera, Stelian Stan

**Affiliations:** Materials Science and Engineering Faculty, “Politehnica” University of Bucharest, 313 Spl. Independentei, 060042 Bucharest, Romania

**Keywords:** high-Si ductile cast iron, solidification, castings, cooling rate, carbides, graphite, graphite nodularity, graphite shape factors, ferrite, pearlite

## Abstract

Ferritic high-Si ductile cast irons replace an unstable mixed ferrite-pearlite matrix with a unique combination of high elongation, strength and hardness (ideal for automotive drive train components) and resistance to oxidation and corrosion at high temperatures (automotive exhaust and turbocharger systems). The present paper analyses the graphite parameters of 4.5%Si, un-inoculated ductile cast iron (4.7%CE, 0.035%Mg_res_) as an effect of the casting section size. The structure is characterized by 10.5–11.2% graphite and 464–975 nodules/mm^2^, at more than 70% ferrite and no carbides, including at 3 mm wall thickness. The lower the wall thickness is, the higher the nodule count is and, consequently, the higher the ferrite amount is. The Roundness Graphite Shape Factor (RSF = 0.65–0.68) illustrates the presence of Slightly Irregular Spheroidal Graphite (Form V ISO 945). There is a big difference between the graphite nodularity evaluated according to ISO 16112:2017 [CGI] (NG_1_ = 79–86%) and according to ISO 945-4-2019 (DI) (NG_2_ = 65.2–74.6%), both of them based on RSF. Graphite Nodularity (NG_3_), calculated with the ISO 945-4-2019 [DI] formula, but replacing RSF with SSF, Sphericity Graphite Shape Factor, has an intermediary position. The higher the imposed minimum RSF or SSF is, the lower the Graphite Nodularity (NG_4_, NG_5_): 80–90% for min. 0.50 (minimum Form IV or Intermediate Graphite), 60–80% for min. 0.60–0.65 (minimum Form V graphite) and 35–70% for min. 0.80 (minimum Form VI graphite). The SSF is more representative than the RSF for Si-alloyed ductile cast iron, so it is recommended to use a graphite nodularity calculus considering SSF instead of the RSF formula (stipulated by ISO 945-4-2019), with SSF replacing RSF.

## 1. Introduction

Ductile cast iron, also known as nodular or spheroidal graphite iron, is a cast iron (ferrous alloy at 3–4%C) with carbon being present in the form of nodular (spheroidal) graphite particles. Nodulizing elements, typically magnesium, with and without an association with rare earth (RE), are used to allow the solidification of the graphite into nodules. Graphite nodules do not form as perfect spheres but with a small or large deviation, depending on a large number of influencing factors, including chemical composition, metallurgical treatments, and solidification conditions. Different graphite shape factors are used to express the deviation of the graphite particle morphology from a sphere as the maximum possible compactness degree. Usually, the Roundness Shape Factor (RSF) is used as a ratio between the graphite particle area and the area of a circle corresponding to the maximum size of the graphite particle (RSF = 1.0 for a sphere). 

In commercial ductile cast iron, depending on the deviation from a sphere, as expressed by RSF values, there are different graphite morphologies defined, according to ISO 945-4-2019: spheroidal (RSF > 0.8, Type VI), slightly irregular spheroidal (RSF = 0.6–0.8, Type V), irregular spheroidal (RSF = 0.45–0.6, Type IV), vermicular/compacted (RSF = 0.1–0.45, Type III), and lamellar (RSF < 0.1, Type I) morphology [1]. These cast irons could also include other unwanted graphite morphologies, such as degenerated form (spiky graphite), exploded graphite, chunky graphite, etc. 

In comparison to steel, while there is no large difference when it comes to tensile strength, ductile iron has greater yield strength. If ductile iron has a static strength comparable to cast steel, this material has greater fatigue strength and ductility than grey (lamellar graphite) cast irons. It is more flexible and more elastic than other cast irons. Generally, as the strength of ductile iron increases, the ductility decreases. Typically, cast iron has better compressive strength than steel.

The ductile iron properties depend on the material structure, with both the metal matrix and the graphite phase parameters acting as important influencing factors. From the metal matrix point of view, the pearlite/ferrite ratio is very important to determine the mechanical properties. The ferrite increasing to the detriment of pearlite (as a result of silicon increase) leads to increased ductility and toughness but to decreased strength and hardness properties. High toughness is particularly important for components that may suffer an impact or for components where a fracture would be catastrophic. 

The graphite particles act as stress raisers, which may prematurely cause localized plastic flow at low stresses and initiate fracture in the matrix at higher stresses. This means that, from a stress concentration point of view, the graphite particles can be considered a defect. It is known that local stresses are minimal with a spherical geometry and increase with another geometry [2]. In ferritic-pearlitic ductile cast iron, it was found that the initiation of microcracks occurred around very irregularly shaped graphite nodules [3]. In the matrix, the softer ferrite gives higher ductility but lower yield strength than pearlite. Graphite morphology plays an important role, and the more the graphite shape deviates from the ideal spherical shape, the lower the ductility and strength [4].

In previous research programs, cast iron samples with different graphite morpho- logy were subjected to a thermal-shock test by cyclic heating (furnace)-cooling (water) procedure [5,6,7]. From time to time, the samples were polished, and the structure was analyzed in unetching conditions. Micro cracks were identified, initiated by conventional defects (micro-inclusions, micro-shrinkages, pores) and graphite particles, which also acted similarly as defects. The increasing number of thermal-shock cycling led to the development of these micro-cracks in crack chains, by the connection of all of these defects, including graphite particles. The microstructures included in Figure 1 illustrate that the micro-cracks mainly occur at the tip of graphite particles, for lamellar (Figure 1a) and vermicular/compacted (Figure 1b) graphite and on the surface of irregularly shaped graphite nodules (Figure 1c). From the nodular graphite type point of view, it is expected that the sensitiveness to micro-cracks initiation increases from type VI (spheroidal) to type V (slightly irregular spheroidal) and, especially, to type IV (irregular spheroidal) morphologies.

In ductile cast iron, silicon (usually as 1.8–3.0%Si content) acts as a graphitizing and ferrite-promoting element, with an important contribution to avoiding carbide formation and a fully ferritic structure target (avoiding pearlite). As a result, silicon increasing in conventional ductile cast irons leads to the improvement of ductility and toughness but to the detriment of strength and hardness. When dissolved in ferrite, silicon favors high strength, hardness, oxidation, and corrosion resistance, at lower ductility, toughness, and thermal conductivity, with graphite as an important influencing factor. High silicon contents (between 3.2 and 4.2%) in the as-cast state are viewed as an opportunity to replace ferritic-pearlitic grades, avoiding potential hardness inhomogeneity. 

It was found that Si-alloying (3.0–6.0%Si) negatively affects the quality of the graphite phase in ductile cast iron. The deviation, using a sphere as a reference of graphite particles, was noticeably increased by Si-alloying (from 2.5 up to 5.5%Si) when a characteristic of the graphite particles appeared to be a larger perimeter, resulting in a large category [IV, V, VI forms, ISO 945] [8,9,10,11]. According to [12,13,14], the 3.5% Si ductile iron showed good nodularity. In contrast, with increasing silicon content to 4.5%, significant graphite degeneracy occurred, with the appearance of chunky graphite, negatively affecting the mechanical properties. Despite that, it was possible to counter the contamination of high-Si ductile iron (3.2%Si, 75 mm thickness Y-IV keel block) with Bi by Ce addition or vice versa, though the form VI graphite would not be achieved [15].

As more and more research programs have shown that, in Si-alloyed ductile cast iron, especially at higher than 3.5%Si content, the graphite phase is negatively affected as regards the compactness degree (at least of Type-V slightly irregular spheroidal graphite presence). The main objective of the present paper is to evaluate the graphite nodularity of 4.5%Si, un-inoculated ductile iron, depending on the graphite shape factors considered and the nodularity calculus formula, when compared to ISO 16112-2017 (CGI) and ISO 945-4-2019 (DI) stipulations, referring to the un-Si alloyed cast irons.

## 2. Materials and Methods

Table 1 summarizes the experimental procedure parameters. The base iron, obtained by electrical melting, is subjected to nodularization treatment by the Tundish-cover technique, with Mg-treatment alloy at a limited content of rare earth elements (RE). Wedge castings (22 mm base, 57 mm height) are obtained in a green sand mold at 1400 W s^1/2^/m^2^ K thermal diffusivity and used for structure analysis. The ferrite/pearlite ratio, the carbide amount, and the graphite phase characteristics are evaluated at different casting section sizes. The graphite characteristics are evaluated with Automatic Image Analysis (OMNIMET ENTERPRISE and analySIS^®^ FIVE Digital Imaging Solutions software) for particles greater than 5 µm in size and 0.59 mm^2^ area of an analyzed field. Different graphite particle size and shape factors, different graphite nodularity formulas, and the relationships between graphite phase parameters are considered. 

In terms of final chemical composition (Table 1), the test ductile cast irons are characterized by 0.035%Mg_res_ and 4.7% carbon equivalent (CE), being included in the hyper-eutectic range (3.3%C and 4.55%Si). The pearlitic sensitiveness factor (Px = 2.3) [16] is strongly affected by silicon alloying (Si/Mn = 20 ratio) and also by the effects of minor elements presence, typical for commercial ductile cast iron. The relatively high antinodulizing influence factor (K = 1.48) [16] also illustrates the commercial ductile cast irons. 

Figure 2 shows the graphite particle size parameters and shape factors considered in the present work. The simple shape factors, such as the Aspect Ratio-AR and Elongation-E, refer to the maximum and minimum size of graphite particles, while the Convexity-Cv compares their Convex (P_c_) and Real (P_r_) perimeters. More complex graphite shape factors include the graphite particle area (A_G_) and one other representative parameter, such as the Maximum Ferret-F_max_ (Roundness Shape Factor-RSF, Equation (1)), the Convex Perimeter-P_c_ (Compactness Shape Factor-CSF, Equation (2)) or the Real Perimeter-P_r_ (Sphericity Shape Factor -SSF, Equation (3)).
Roundness Shape Factor: RSF = 4 A_G/_π F_max_^2^(1)
Compactness Shape Factor: CSF = 4 πA_G_/P_c_^2^(2)
Sphericity Shape Factor: SSF = 4 π A_G/_P_r_^2^(3)

Graphite Nodularity (NG) usually refers to the defined nodular (spheroidal) graphite rate compared to the total graphite particles in the analyzed structure, regarding their number or area amount, respectively. The most important problem in this investigation is to define the nodular (spheroidal) graphite morphology considered, as it varies within a large range, from lamellar (maximum/minimum ratio < 0.1) to a sphere shape (maximum/minimum size = 1.0). In this respect, the different international standards referring to cast irons stipulate different formulas for industrial applications.

Figure 3 illustrates the most important terms, included in ISO 16112:2017, referring to the Compacted Graphite Cast Iron (CGI) (Figure 3a) [17], and in ISO 945-4-2019, referring to the ductile (nodular/spheroidal graphite) cast iron (Figure 3b) [1]. Both of them are based on the area of the graphite particles and their Roundness Shape Factor (RSF), including the graphite particles area (A_G_) and Maximum Ferret (F_max_), but at the different ranges considered. According to the CGI-ISO Standard, graphite particles are defined as nodules (RSF = 0.625–1.0), intermediates (RSF = 0.525–0.625), and vermicular/compacted (RSF < 0.525). ISO 945-4-2019, applied to ductile cast irons, refers to spheroidal (RSF = 0.45–1.0), vermicular/compacted (RSF = 0.10–0.45) and lamellar (RSF < 0.10), with three classes of spheroidal graphite, namely spheroidal (RSF ≥ 0.80), slightly irregular spheroidal (RSF = 0.60–0.80) and irregular spheroidal (RSF = 0.45–0.65), respectively.

According to ISO 16112:2017-CGI (Figure 3a), the standard considers all of the graphite particles defined as nodules (RSF > 0.625) and 50% of the graphite particles included in the intermediate class (RSF = 0.525–0.625). ISO 945-4-2019-DI considers all of the graphite particles defined by spheroidal graphite (RSF ≥ 0.80) and 90% of the slightly irregular graphite particles (RSF = 0.60–0.80). A comparison of the respective graphite nodularity, defined by these two standards, is illustrated in Figure 4. 

## 3. Results and Discussion

Typical microstructures of the wedge casting at the 3, 9, and 15 mm section sizes, without and with Nital etching, are shown in Figure 5. For all of the considered solidification cooling rates induced by wall thickness variation, the un-inoculated, 4.5%Si ductile cast iron shows a ferrite–pearlite matrix (at different ferrite/pearlite ratios), without free carbides and with mainly nodular (spheroidal) graphite phase, but with size and morphology.

Parameters depend on the cooling rate. It is known that the lower the casting section size is, the higher the solidification cooling rate. Generally, the solidification cooling rate increase affects both primary (eutectic) and secondary (eutectoid) structures regarding the constituents’ formation and their parameters. For conventional iron castings, it is expected to increase the sensitivity to free carbides formation (instead of graphite) during eutectic solidification, and pearlite (instead of ferrite) during eutectoid transformation, respectively.

Meanwhile, 4.5%Si alloyed ductile cast iron appears to have some peculiar structure formation parameters from both graphite and metal matrix point of view. According to Figure 5, the decrease of the casting wall thickness from 15 to 3 mm, led not only to the expected increase in the nodule count (from 464 up to 975 Nodules/mm^2^) but also to the increase in the ferrite amount (from 70 up to 100%). Ferrite formation during the eutectoid reaction is controlled by carbon diffusion from the austenite matrix to the existent graphite particles. The higher the carbon diffusion capability, the higher the amount of ferrite formed. Carbon diffusion is mainly sustained by higher silicon content in the austenite (lower carbon stability) and the shorter distance between graphite particles (such as for higher graphite particle count). Still, it also depends on the diffusion time available (the higher the cooling rate, the lower the time available for carbon diffusion). For this reason, it is expected to obtain a lower ferrite/pearlite ratio by decreasing the casting section size, but this situation appears to be typical only for un-silicon alloyed ductile cast irons. The decreased carbon stability in the austenite structure (by Si-alloying) and the decreased distance between nodules (higher nodule count) favored the carbon diffusion, able to form ferrite, despite the decrease of the eutectoid reaction time, due to the decrease of the casting size, respectively.

It appears that, in Si-alloyed ductile cast irons, the major increase of the nodule count could be a solution for obtaining free carbides and pearlite structure, including in thin wall casting solidification conditions, such as 3 mm casting section size. Similarly [18], for EN-GJS-SiMo45-6 ductile iron, it was found that the increase of the nodule count by reducing the casting section size resulted in reducing the carbon and molybdenum segregation to the intercellular regions and, hence, in reducing the number of intercellular precipitates. The results suggest that pearlite and carbides are related to segregations during solidification rather than to cooling rates at the eutectoid temperature.

Figure 6 shows the strong influence of the wedge casting wall thickness (lower section size, higher solidification cooling rate) on the nodule count, in the present test conditions, for 4.5%Si-alloyed ductile cast iron and from 3 up to 15 mm casting section size. More than 900 nodules per square millimeter resulted in the smallest casting section size (3 mm), 600–700 for the 6 mm casting section size, and 450–500 for the 9–15 mm casting section size. 

Figure 7 illustrates the influence of the wedge-casting wall thickness on the graphite nodules size rate distribution. Generally, the decrease in the casting section size (solidification cooling rate increases) favors the increase of the rate of lower graphite nodule size. More than 90% of graphite nodules are less than 30 μm, with a visible peculiar position of 3 mm section size: 65–70% less than 15 μm size and 30–35% at a rate of 15–30 μm nodules size. The second values group brings together 6–15 mm casting section sizes, with 30–40% nodules less than 15 μm and 50–60% nodules at 15–30 μm size, respectively. 

The amount of the graphite particles (G) and their convex (P_c_) and real (P_r_) perimeter (see Figure 2) are also influenced by the wedge-casting wall thickness (Figure 8). If the graphite amount is less depending on the wall thickness (G = 10.5–11.2%), the values of both the convex and real perimeters are visibly influenced: less than 40 μm for the 3 mm casting section size, 45–50 μm for the 6 mm section size, and 50–60 μm for the 9–15 μm section size. The highest values of the considered graphite particles are obtained for a 12 mm wedge casting section size, much more than for the thickest section due to the end effect (at 15 mm section). A higher level of real perimeter than the convex perimeter resulted in the entire range of wedge casting section size considered.

The graphs included in Figure 9 show the evolution of other graphite shape factors (defined by Figure 2) as an influence of the wedge-casting wall thickness. As general values, the considered graphite shape factors fall within a narrow field, for the entire range of the wedge casting section size: Aspect Ratio AR = 1.32–1.39, Elongation E = 1.37–1.48, Convexity Cv = 0.91–0.92, Sphericity SSF = 0.79–0.82, Roundness RSF = 0.65–0.68. The Roundness Graphite Shape Factor (RSF), which is considered (by the international standard ISO 945-4-2019) to characterize the representative graphite morphologies in cast irons (Figure 3b), suggests a fall within the slightly irregular spheroidal graphite (Form V ISO 945), in the 4.5%Si ductile cast iron test. For RSF < 0.7, it practically is at the lowest level of the compactness degree defined by this standard (RSF = 0.6–0.8). 

Despite the fact that the same graphite shape factor (RSF) is used, there is a big difference between the graphite nodularity evaluated according to ISO 16112:2017 [CGI] (NG_1_ = 79–86%) and according to ISO 945-4-2019 (DI) (NG_2_ = 65.2–74.6%). 

Graphite nodularity was basically calculated by considering the Roundness Shape Factor (RSF), according to ISO 16112:2017, applied for the compacted graphite cast iron (NG_1_-Figure 3a and Figure 4) and to ISO 945-4-2019, applied for the ductile cast iron (NG_2_-Figure 3b and Figure 4). The results obtained are illustrated in Figure 10, with the wedge-casting wall thickness as a possible

By using the graphite nodularity NG_2_ formula (Figure 3b), while replacing the Roundness Shape Factor (RSF) with the Sphericity Shape Factor (SSF), the graphite nodularity resulted: NG_3_ = [(∑A_NG(SSF≥0.8)_ + 0.9∑A_NG(SSF=0.6–0.80)_)/∑A_tot_]·100 (%). Its evolution with the wedge casting section size is also illustrated in Figure 10. Practically having the same appearance as the NG_2_ line, graphite nodularity NG_3_ has an intermediate position: very close to NG_1_ for thin wall thickness (3–6 mm), close to 9 mm thickness, and at an equidistant position of 12–15 mm wall thickness.

A specific graphite nodularity was also calculated, referring to a minimum imposed graphite shape factor (RSF or SSF), in the 0.5–0.8 range: ${\mathrm{NG}}_{4}=\frac{\Sigma A_{particles\;\left(RSF\right)}}{\Sigma A_{tot}}\cdot 100$ [%] and ${\mathrm{NG}}_{5}=\frac{\Sigma A_{particles\;\left(SSF\right)}}{\Sigma A_{tot}}\cdot 100$ [%] (Figure 11).

In other words, in the graphite nodularity calculus, the area considered was the area of all of the graphite particles with shape factors RSF or SSF at a minimum imposed level (0.50, 0.60, 0.65, and 0.80) relative to the total area of the present graphite particles. The higher the graphite shape factor level, the lower the resulting graphite nodularity for both NG_4_ and NG_5_ expressions. If the minimum presence of Form IV (intermediate or irregular spheroidal) is accepted (minimum 0.50 for RSF or SSF), the graphite nodularity is at the highest level, 80–90%, respectively. Graphite nodularity decreases to 60–80% if the minimum considered graphite morphology is Form V (slightly irregular spheroidal graphite), typically for the 0.6–0.8 graphite shape factor. The lowest graphite morphology corresponds to the highest considered graphite shape factor level (minimum 0.8), characteristic for Form VI (regular spheroidal graphite): 55–70% for SSF = min. 0.80 and less than 40% for RSF = min. 0.80, respectively. In this case, the highest difference is also registered between the graphite nodularity levels, obtained using the graphite phase’s RSF or SSF parameters. 

Figure 12 shows the influence of the minimum imposed graphite shape factors (RSF, SSF) on the graphite nodularity NG_4_ and NG_5_ and the wedge-casting wall thickness. This figure shows not only the decrease of the graphite nodularity by the increase of the claims on the graphite phase compactness degree but also the increasing difference expressed by the RSF or SSF parameters contribution. The difference is less visible for a min. 0.5 value, then it continually increases between NG_4_ and NG_5_, up to the minim. 0.8 level of graphite shape factor (from 10 up to 35% graphite nodularity difference).

Micro-structures obtained in unetched wedge casting samples, at 3 and 15 mm wall thickness, are compared in Figure 13 regarding the graphite phase aspect and also regarding the representative graphite parameters values obtained (nodule count and nodularity). The solidification cooling rate, expressed by the wedge casting section size (higher section size, lower cooling rate), strongly affects the graphite phase, especially as nodule count, which was cut by half at 15 mm compared to 3 mm wall thickness. The wall thickness increase also led to an increase in the rate of larger graphite particles and the formation of compacted graphite morphology. Both of these registered effects are generally expected in ductile iron castings. As the microstructures suggest, the calculated graphite nodularity level is also expected to be negatively affected by the increase in the casting wall thickness. However, the graphite nodularity, expressed by NG_1_ and NG_2,_ does not appear to sustain it: NG_1_ = 81.5% vs. 81.7% and NG_2_ = 65.9% vs. 65.3%, 

For 3 mm vs. 15 mm casting section size solidification, only the NG_3_ graphite nodularity calculated illustrates the visible decrease of the nodularity (79.6% vs. 73.0%). It appears that the Sphericity Shape Factor (SSF), involving the graphite particles area and the real perimeter, is more representative than the Roundness Shape Factor (RSF), involving the graphite particles area and the maximum ferret, for Si-alloyed ductile cast iron. Therefore, using a graphite nodularity calculus is recommended considering the SSF instead of the RSF formula (stipulated by ISO 945-4-2019), with SSF replacing RSF. 

## 4. Conclusions

The present paper evaluates the graphite nodularity in 4.5%Si, commercial ductile cast iron (0.035%Mg_res_, 4.5%Si, 4.7%CE, Px = 2.3, K = 1.48), and un-inoculated conditions, solidified in thin wall castings (up to 20 mm section size) via green sand mould as the effect of Si alloying, the graphite particles shape factors, the graphite nodularity calculus formula, and the wedge casting wall thickness variation. The following conclusions could be drawn:

*The structure is characterized by 10.5–11.2% graphite (43% nodules at max 15 μm and 50% at 15–30 μm) and 464–975 nodules/mm^2^, at more than 70% ferrite and no carbides, including at 3 mm wall thickness. The lower the wall thickness, the higher the nodule count, and the higher the ferrite amount. The high cooling rate is generally favorable for carbides and pearlite formation.

*The Roundness Shape Factor (RSF = 0.65–0.68), involving the graphite particle area (A_G_) and its maximum size (F_max_), illustrates the presence of slightly irregular spheroidal graphite (Form V ISO 945), characterized by a higher real perimeter, and positioned at the lower part of this field (RSF = 0.60–0.80, ISO 945-4-2019-DI). Other shape factors also sustain the lower quality of this spheroidal graphite: AR = 1.32–1.39, E = 1.37–1.48, Cv = 0.91–0.92, SSF = 0.79–0.82.

*There is a big difference between the graphite nodularity evaluated according to ISO 16112:2017 [CGI] (NG_1_ = 79–86%) and according to ISO 945-4-2019 (DI) (NG_2_ = 65.2–74.6%), both of them based on the Roundness Shape Factor (RSF). The same evolution of the NG_1_ and NG_2_ was registered with the wall thickness variation. 

*Graphite Nodularity (NG_3_), calculated with the ISO 945-4-2019 [DI] formula, but by replacing the RSF with the Sphericity Shape Factor (SSF), involving the graphite particle area (A_G_) and its real perimeter (P_r_), has an intermediary position: very close to NG_1_ for a thin wall thickness (3–6 mm), close for a 9 mm thickness, and at an equidistant position for a 12–15 mm wall thickness. 

*The higher the imposed minimum RSF or SSF graphite shape factors, the lower the graphite nodularity [NG_4_, NG_5_]: 80–90% for both the RSF and SSF = min. 0.50 (minimum Form IV or Intermediate Graphite), 60–80% for RSF and SSF = min. 0.60–0.65 (minimum Form V), and 35–70% for RSF and SSF = min. 0.80 (minimum Form VI). 

*As a general conclusion, for high-Si DI (mainly > 4%Si), the Graphite Shape Factor (SSF), involving the graphite particle area (A_G_) and the real perimeter (P_r_), is more representative than the RSF Shape Factor, involving the graphite particle area (A_G_) and its maximum size (F_max_).

*In high Si-Ductile cast irons, it is recommended to use a graphite nodularity calculus considering the SSF instead of the RSF formula (stipulated by ISO 945-4-2019), with SSF replacing RSF:NG = [(∑A_NG(SSF≥0.80)_ + 0.9∑A_NG(SSF=0.60–0.80)_)/∑A_tot_]·100 (%) [SSF = 4 π A_G_/P_r_^2^]

## Figures and Tables

**Figure 1 materials-15-07685-f001:**
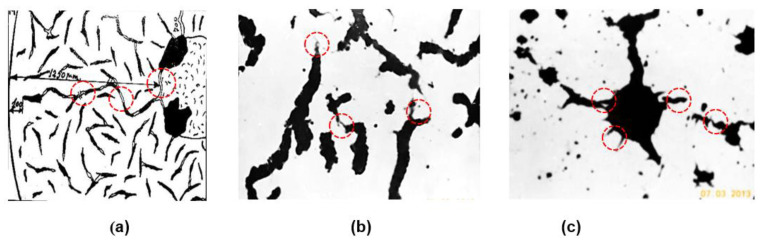
Micro-cracks developed on the tip of the lamellar (**a**) and vermicular/compacted (**b**) graphite particles and on the surface of irregularly shaped graphite nodules (**c**) as a result of the thermal-shock test. The red circle shows the micro-crack initiation.

**Figure 2 materials-15-07685-f002:**
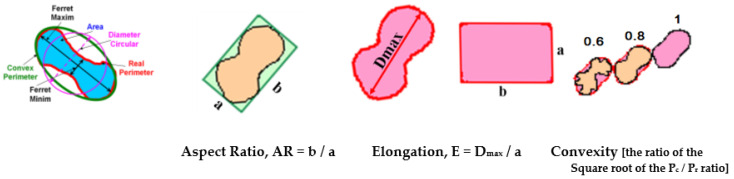
Graphite particle size parameters and shape factors.

**Figure 3 materials-15-07685-f003:**
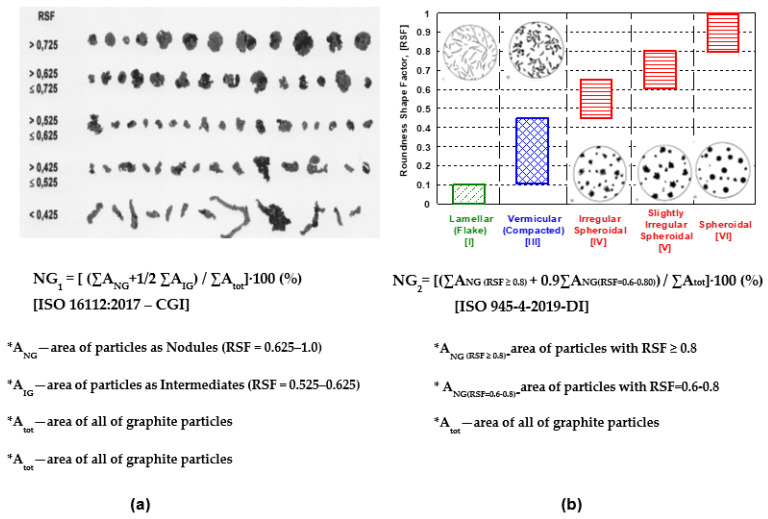
Calculus of the Graphite Nodularity (NG) depending on the Roundness Shape Factor (RSF) according to ISO 16112-2017-CGI (**a**) and ISO 945-4-2019-DI (**b**).

**Figure 4 materials-15-07685-f004:**
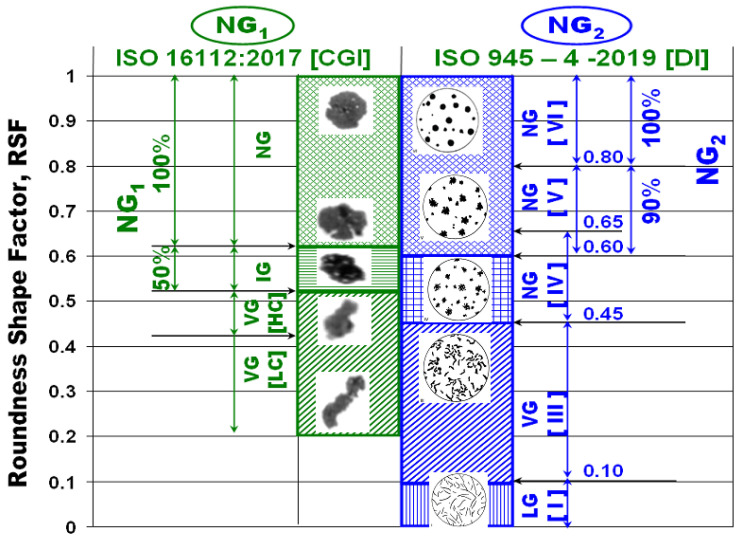
Graphite morphologies and Roundness Shape Factor (RSF) involved in the graphite nodularity calculus, according to ISO 16112-2017-CGI (NG_1_) and ISO 945-4-2019-DI (NG_2_).

**Figure 5 materials-15-07685-f005:**
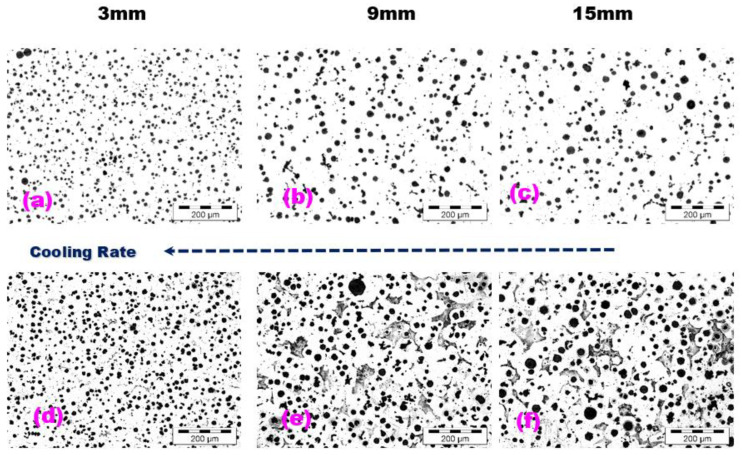
Influence of the wedge casting wall thickness on the structure of the 4.5%Si Un-inoculated ductile cast iron (**a**,**d**) 3 mm; (**b**,**e**) 9 mm; (**c**,**f**) 15 mm; (**a**–**c**) un-etching; (**d**–**f**) Nital etching.

**Figure 6 materials-15-07685-f006:**
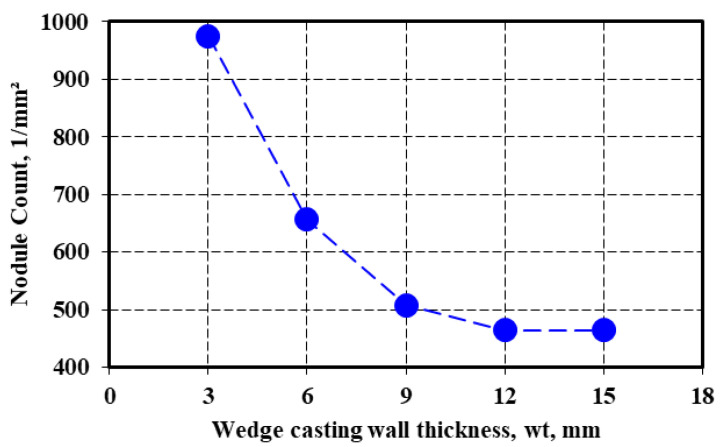
Influence of wedge casting wall thickness on the nodule count.

**Figure 7 materials-15-07685-f007:**
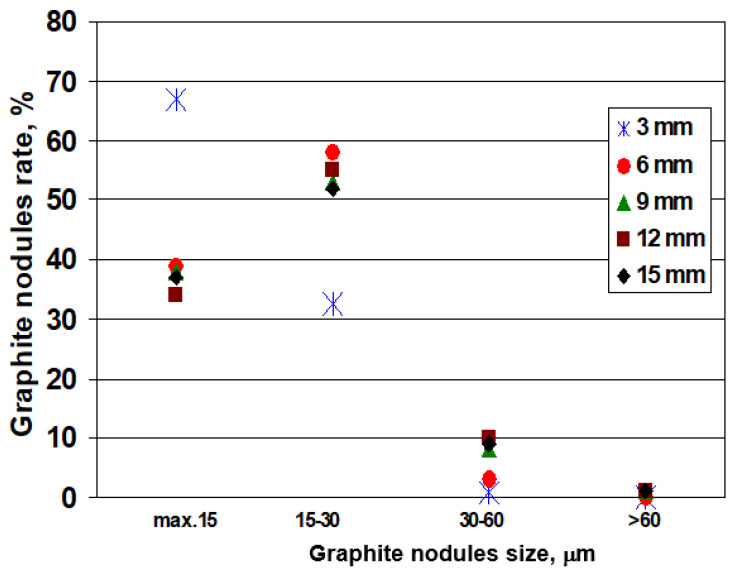
Influence of wedge casting wall thickness on the graphite nodules size rate distribution.

**Figure 8 materials-15-07685-f008:**
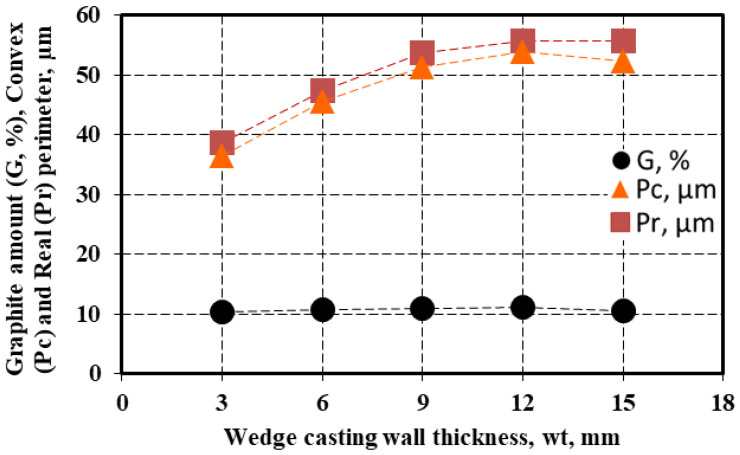
Influence of the wedge casting wall thickness on the average graphite amount (G), convex perimeter (P_c_), and real perimeter (P_r_) of graphite particles.

**Figure 9 materials-15-07685-f009:**
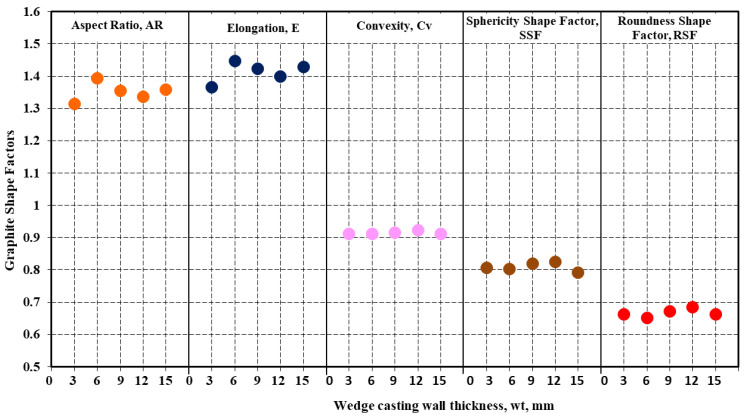
Graphite shape factors as an influence of the wedge casting wall thickness influencing factor.

**Figure 10 materials-15-07685-f010:**
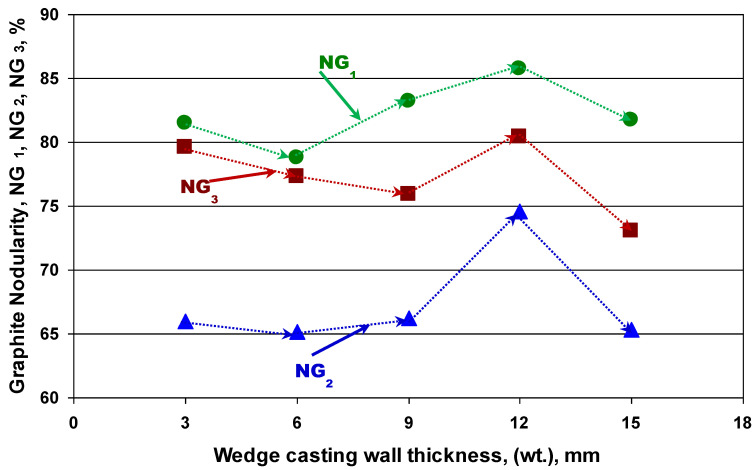
Graphite nodularity calculated using the graphite Roundness Shape Factor (RSF), according to ISO 16112:2017–CGI (NG_1_) and ISO 945-4-2019-DI (NG_2_), and by replacing the RSF with the Graphite Sphericity Shape Factor (SSF) according to ISO 945-4-2019-DI formula (NG_3_).

**Figure 11 materials-15-07685-f011:**
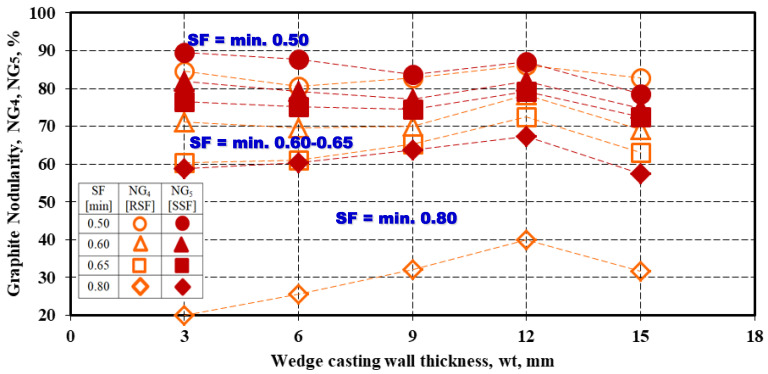
Graphite nodularity NG_4_ [RSF] and NG_5_ [SSF] for the minimum imposed values for Graphite Shape Factors (SF), RSF, and SSF.

**Figure 12 materials-15-07685-f012:**
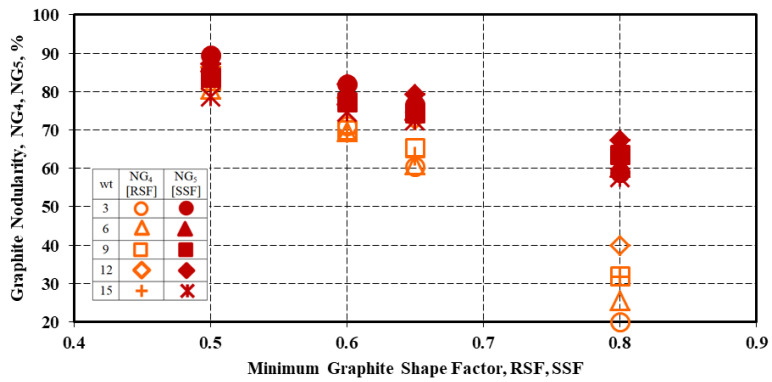
Influence of the minimum imposed Graphite Shape Factors (RSF, SSF) on Graphite. Nodularity NG_4_ and NG_5_, as an influence of the wedge casting wall thickness.

**Figure 13 materials-15-07685-f013:**
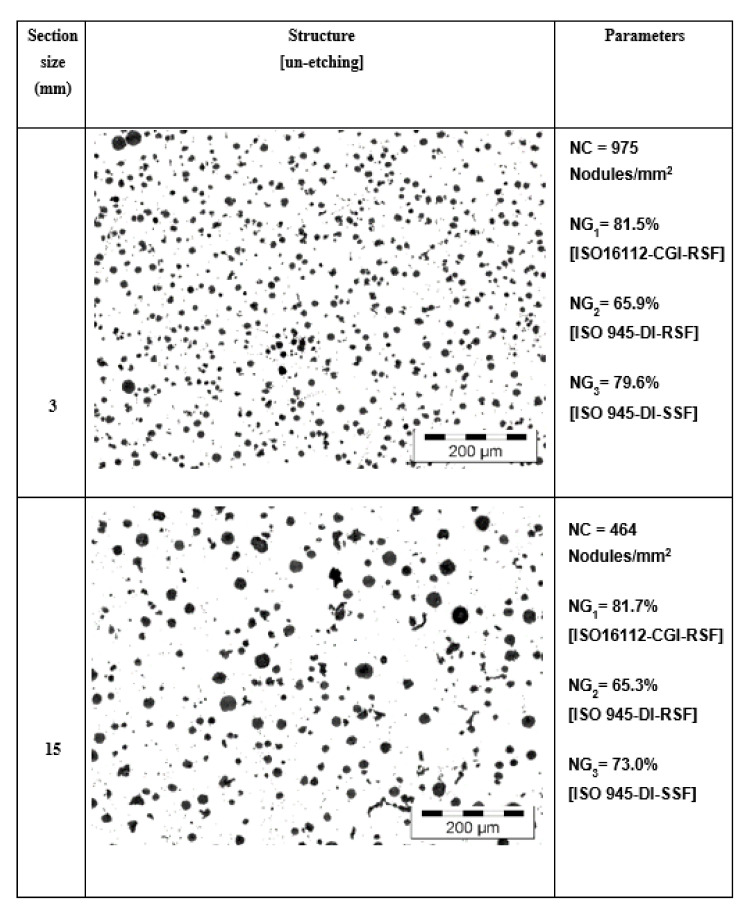
Comparison of the microstructures, the nodule count (NC), and the calculated graphite nodularity [NG_1_, NG_2_, NG_3_] for the 3 and 15 mm wedge casting wall thickness.

**Table 1 materials-15-07685-t001:** Experimental procedure parameters.

Nr.	Step	Characteristics
1	Melting	Coreless induction furnace, graphite crucible, 10 kg, 8000 Hz; Pig iron, cast iron scrap, recarburizer; 1525 °C superheating, 5 min holding
2	Nodularization	Tundish cover technique, 10 kg ladle, 1500 °C temperature treatment; 1.5 wt.% FeSiCaMgRE [wt.%: 5.99Mg, 1.0Ca, 0.26RE, 0.91Al, 44.7Si, bal Fe]
	Casting	Wedge castings (22 mm base, 57 mm height) Green sand mould [SM]; 1400 W s^1/2^/m^2^ K thermal diffusivity
4	Structure Analysis	The graphite characteristics are evaluated with Automatic Image Analysis (OMNIMET ENTERPRISE and analySIS^®^ FIVE Digital Imaging Solutions software) for particles greater than 5 µm and 0.59 mm^2^ as the size area of an analyzed field. * Carbides and ferrite/pearlite ratio * Different graphite particles size and shape factors * Different graphite nodularity formulas * Relationships between graphite phase parameters
5	Final Chemical Composition	(wt.%): 3.3C, 4.55Si, 0.22Mn, 0.04P, 0.01S, 0.035Mg, 0.0004Ce, 0.0061La, 0.0038Ca; CE = 4.7%; Control Thielmen Factors [16]: Px = 2.3; K = 1.48 CE = % C + 0.3(% Si + % P) + 0.4(% S) − 0.027(% Mn) Px = 3(% Mn) − 2.65(% Si − 2) + 7.75(% Cu) + 90(% Sn) + 357(% Pb) + 333(% Bi) + 20.1(% As) + 9.60 (% Cr) + 71.7(% Sb) K = 4.4(% Ti) + 2.0(% As) + 2.4(% Sn) + 5.0(% Sb) + 290(% Pb) + 370(% Bi) + 1.6(% Al)

(*) The structure analysis objectives.

## Data Availability

Not applicable.

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
