# Peer review of "Graphite Nodularity Evaluation in High-Si Ductile Cast Irons"

_materials, 2022, doi:10.3390/ma15217685_

Round 1
Reviewer 1 Report
Manuscript ID: Materials-1979022
In the submission titled “Graphite Nodularity Evaluation in High-Si Ductile Cast Irons”, Isahak et al. reported the study for the evaluation of graphite nodularity in commercial ductile iron with 4.5% Si compound. First of all, I suggest that the authors should improve the clarity of some figures such as Figure 5 and Figure 13 and the main text. Due to the lack of clarity, it was hard to understand. Maybe, the potential readers will suffer this issue. Especially, in the section of Conclusions, the authors listed the key findings without any highlighting and conciseness. In addition, I felt that the citing literatures are not enough for providing the background and significance of this work. I strongly recommend that the author should revise the main text and figure to improve the readability.
Author Response
Reviewer #1:
- In the submission titled “Graphite Nodularity Evaluation in High-Si Ductile Cast Irons”, Isahak et al. reported the study for the evaluation of graphite nodularity in commercial ductile iron with 4.5% Si compound.
Answer: The mentioned authors (Isahak et al.) do not refer to the present paper
- First of all, I suggest that the authors should improve the clarity of some figures such as Figure 5 and Figure 13 and the main text. Due to the lack of clarity, it was hard to understand. Maybe, the potential readers will suffer this issue.
Answer: Figure 5 and Figure 13 have been modified, to increase their clarity.
- Especially, in the section of Conclusions, the authors listed the key findings without any highlighting and conciseness.
Answer: The paper was revised in this respect.
- I strongly recommend that the author should revise the main text and figure to improve the readability
Answer: The modifications were made.
- In addition, I felt that the citing literatures are not enough for providing the background and significance of this work.
Answer: There were considered representative literature sources to sustain the mentioned main objective of the present work.
Reviewer 2 Report
The structure of this paper is clear, and the research results can be accurately obtained. However, there are still some problems with errors, poor articulation, poor interpretation and misstatement. The authors should improve his or her English and at the same time be more careful in writing the paper to avoid many minor flaws. There are mainly the following points:
1. More information from Figure 7 is needed.
2. Explain clearly which is the size of Slightly Irregular Spheroidal Graphite?
3. Explain the final shape factor in more detail.
4. What is the influence of the process of casting on the graphite structure such as graphitizing and defects? Some references can be added for analyzing the micro structure of the formed graphite.
Zhao J, Liu Y, Liu D, et al. The Tribological Performance of Metal-/Resin-Impregnated Graphite under Harsh Condition. Lubricants, 2021, 10(1): 2.
Mussa A, Krakhmalev P, Bergström J. Wear mechanisms and wear resistance of austempered ductile iron in reciprocal sliding contact. Wear, 2022, 498: 204305.
5. The motivation of this manuscript should be demonstrated clearly. What is the difference between this manuscript and the below paper? That should be discussed as well.
Anca D E, Stan I, Riposan I, et al. Graphite Compactness Degree and Nodularity of High-Si Ductile Iron Produced via Permanent Mold versus Sand Mold Casting[J]. Materials, 2022, 15(8): 2712.
6. in Page 1, Line 15-16.
Change “higher ferrite amount” to “the higher ferrite amount is”
7. in Page 1, Line 32.
Change “Ductile Iron” to “Ductile cast iron”
8. in Page 9, Line 357.
Change “regards” to “regarding”.
Author Response
- Reviewer #2:
The structure of this paper is clear, and the research results can be accurately obtained. However, there are still some problems with errors, poor articulation, poor interpretation and misstatement. The authors should improve his or her English and at the same time be more careful in writing the paper to avoid many minor flaws. There are mainly the following points:
- a) More information from Figure 7 is needed.
Answer: More information added.
- Explain clearly which is the size of Slightly Irregular Spheroidal Graphite?
Answer: the size of Slightly Irregular Spheroidal Graphite particles resulted by the
scale of each microstructure and Figure 7, with added information.
- Explain the final shape factor in more detail.
Answer: the present paper considered 6 shape factors to characterize the graphite particles. The last mentioned is SSF-Sphericity Shape Factor, what is mainly
characterized by the use of the real graphite particle perimeter. This size parameter was particularly found to be affected by silicon alloying of ductile cast irons. A mention was added in the manuscript.
- What is the influence of the process of casting on the graphite structure such as graphitizing and defects? Some references can be added for analyzing the micro structure of the formed graphite. Zhao J, Liu Y, Liu D, et al. The Tribological Performance of Metal-/Resin-Impregnated Graphite under Harsh Condition. Lubricants, 2021, 10(1): 2. Mussa A, Krakhmalev P, Bergström J. Wear mechanisms and wear resistance of austempered ductile iron in reciprocal sliding contact. Wear, 2022, 498: 204305.
Answer: The present paper refers to the graphite formation in high-Si ductile cast iron (4.5%Si), documented to have a specific solidification pattern, comparing to the conventional, un-Si alloyed ductile cast irons (< 3%Si). Thank you for the recommended sources, they will be useful in other research works, but not now, as both of them refer to other silicon range in ductile cast irons.
- The motivation of this manuscript should be demonstrated clearly. What is the difference between this manuscript and the below paper? That should be discussed as well. Anca D E, Stan I, Riposan I, et al. Graphite Compactness Degree and Nodularity of High-Si Ductile Iron Produced via Permanent Mold versus Sand Mold Casting[J]. Materials, 2022, 15(8): 2712.
Answer: In the mentioned work, the graphite nodularity was calculated according to the Roundness Shape Factor (RSF, involving maximum graphite particle size, Fmax) and nodular/spheroidal definition according to the actual applied international standard ISO 945-4-2019: RSF ≥ 0.6 to 1.0. It was considered the total area of graphite particles with RSF ≥ 0.8 and 90% of the total area of graphite particles with RSF=0.6-0.8.
This technique was found to be useful and generally used in present in the worldwide cast iron foundry industry, referring to the un-Si-alloyed ductile cast irons (< 3%Si). In the mentioned work, by this way were compared obtained structure in sand mould and metal mould solidification.
As it was documented that Si – alloying negative affected the compactness degree of graphite particles (mainly as increased real perimeter), it was considered necessary to find a more accurate technique to evaluate graphite nodularity in these ductile cast irons. And the present paper proposes such possibility.
- in Page 1, Line 15-16. Change “higher ferrite amount” to “the higher ferrite amount is”
Answer: The modification was made.
- in Page 1, Line 32 Change “Ductile Iron” to “Ductile cast iron”
Answer: The modification was made.
- in Page 9, Line 357.Change “regards” to “regarding”.
Answer: The modification was made.
Reviewer 3 Report
Comments:
1) What is the significance of “Wedge Casting”? Why is it preferred for this research work?
2) In general, Sand Casting process results in poor yield, quality and less density. How this process can be used for performing experiments for critical applications?
3) The findings reveal that wall thickness plays a major role in defining the physical and microstructural properties of the component. What should be the threshold value for critical applications. Because the wall thickness also defines the strength of the component.
4) The findings are more theoretical rather than real time application orientation. Authors need to justify this fact.
5) Authors may suggest the way to maintain the prescribed values of certain parameters while manufacturing the components through this casting process.
6) Very old references may be removed and some more recent references may be added.
Author Response
III. Reviewer #3:
- What is the significance of “Wedge Casting”? Why is it preferred for this research work?
Answer: Wedge Casting is cast test specimen used for assessment of chill depth, particularly for inspection of cast iron, defined by Standard A367-85: Standard Test Methods of Chill Testing of Cast Iron; American Society for Testing of Materials: West Consho55, PA, USA, 2000; pp. 151-154. This wedge test type is the most use test sample in the cast iron industry worldwide. Information added in the manuscript.
- In general, Sand Casting process results in poor yield, quality and less density. How this process can be used for performing experiments for critical applications?
Answer: For the mentioned negative results, the Sand casting process is less used in the Al-alloys castings production, where permanent moulds, die castings techniques, etc are preferred. But, for iron castings production, sand mould techniques are large used, at different variants as binder systems.
- The findings reveal that wall thickness plays a major role in defining the physical and microstructural properties of the component. What should be the threshold value for critical applications. Because the wall thickness also defines the strength of the component.
Answer: the limited wall thickness of the selected wedge casting test refers mainly for the thin wall castings production, more and more attractive in the foundry industry worldwide
- The findings are more theoretical rather than real time application orientation. Authors need to justify this fact.
Answer: you are not right. The main conclusion of the present work is the necessity to use other proposed formula to calculate the graphite nodularity in High-Si ductile cast irons production, as the actual applied international standard ISO 945-4-2019 stipulation is useful for un-Si-alloyed irons, but not for Si-alloying negative effects on the spheroidal graphite compactness degree. Our proposal is to consider other formula for these materials.
- Authors may suggest the way to maintain the prescribed values of certain parameters while manufacturing the components through this casting process.
Answer: it was not the objective of the present work.
- Very old references may be removed and some more recent references may be added.
Answer: 12 bibliographic sources (out of 19) have published after 2018. The ‘very old sources’, what sustain Figure 1 structures, are representative for the mechanism of micro-cracks formation as effects of spheroidal graphite particles irregularities, as it was found that this morphology mainly characterizes the graphite particles in High-Si ductile cast irons.
Obs. All of the recorded changes were marked by colour style in the revised manuscript.